# Quantum magnetic *J*-oscillators

Jingyan Xu [1,2,3], Raphael Kircher [1,2,3], Oleg Tretiak [1,2,3], Dmitry Budker [1,2,3,4] & Danila A. Barskiy [1,2,3,5] ✉

Zero-field nuclear magnetic resonance (NMR) offers magnet-free access to nuclear spin-spin (scalar *J*) couplings, which define an intrinsic, molecule-specific frequency scale. However, the transient nature of zero-field NMR signals constrain spectral resolution and frequency stability. Here we introduce quantum *J*-oscillators that exploit *J*-couplings in molecules to produce phase-coherent continuous oscillations. Operated in zero magnetic field and driven by digital feedback, they generate sub-hertz to a few tens of hertz frequencies. In a proof-of-principle experiment on [$^{15}$N]-acetonitrile, the oscillator achieves a 340 $\mu$Hz linewidth over 3600 s, more than two orders of magnitude narrower than in conventional zero-field NMR. This methodology may facilitate precision measurements of *J*-coupling constants and enables discrimination of molecules whose zero-field NMR spectra are otherwise difficult to resolve. In addition, the combination of strongly coupled spin systems and programmable feedback turns *J*-oscillators into a compact tabletop platform for exploring nonlinear spin dynamics, including chaos and dynamical phase transitions. By uniting high-resolution spectroscopy and controllable quantum dynamics in a single, magnet-free setup, *J*-oscillators open new opportunities for applications where ultraprecise frequency references or molecular fingerprints are required.

Masers and lasers have revolutionized science and technology, finding applications in fields as diverse as telecommunications, medical diagnostics, astronomy, precision measurements, and fundamental physics[1–7]. Both technologies harness coherent electromagnetic radiation amplified through the process of stimulated emission[8], a phenomenon that traditionally requires achieving population inversion between quantized energy states (i.e., a higher-energy state has to be more populated than a lower-energy state)[9].

Conventional masers have been realized across diverse physical systems, including atomic beams[1,10], gases[11,12], and solid-state systems[13–17]. These devices typically operate in the GHz frequency range, achieving population inversion via various mechanisms. Recent advances extended these principles to the so-called "rasers" generating kHz-to-MHz frequencies using nuclear spins[18]. To create the required population inversion, such low-frequency rasers rely on hyperpolarization approaches such as spin-exchange optical

pumping[12], dynamic nuclear polarization[19], photochemical polarization transfers[20], and parahydrogen-based techniques[18,21–24]. Additionally, their emission amplification is typically triggered by "radiation damping"[25] stemming from inductive coupling of polarized nuclear spins with detection coils. However, these rasers exhibit intrinsic limitations as they operate on Zeeman-split levels. Since the frequencies of these levels depend on the applied (bias) magnetic field, they are susceptible to magnetic field drifts, limiting their long-term stability and reproducibility.

In this work, we introduce zero-field quantum oscillators operating at frequencies from near-DC to tens of hertz, overcoming limitations of conventional rasers that rely on Zeeman-split levels. Unlike those, our oscillators function without a bias field, exploiting intrinsic nuclear spin-spin scalar interactions (*J*-couplings) within molecules[26,27]. Importantly, they utilize $\Delta m = 0$ transitions[27], with the quantization axis along the measurement axis (Fig. 1A,B). Because these transition

[1]Helmholtz Institute Mainz, Mainz, Germany. [2]GSI Helmholtzzentrum fur Schwerionenforschung, Darmstadt, Germany. [3]Institute of Physics, Johannes Gutenberg-Universität, Mainz, Germany. [4]Department of Physics, University of California, Berkeley, CA, USA. [5]Frost Institute for Chemistry and Molecular Science, Department of Chemistry, University of Miami, Coral Gables, FL, USA. ✉e-mail: barskiy@miami.edu

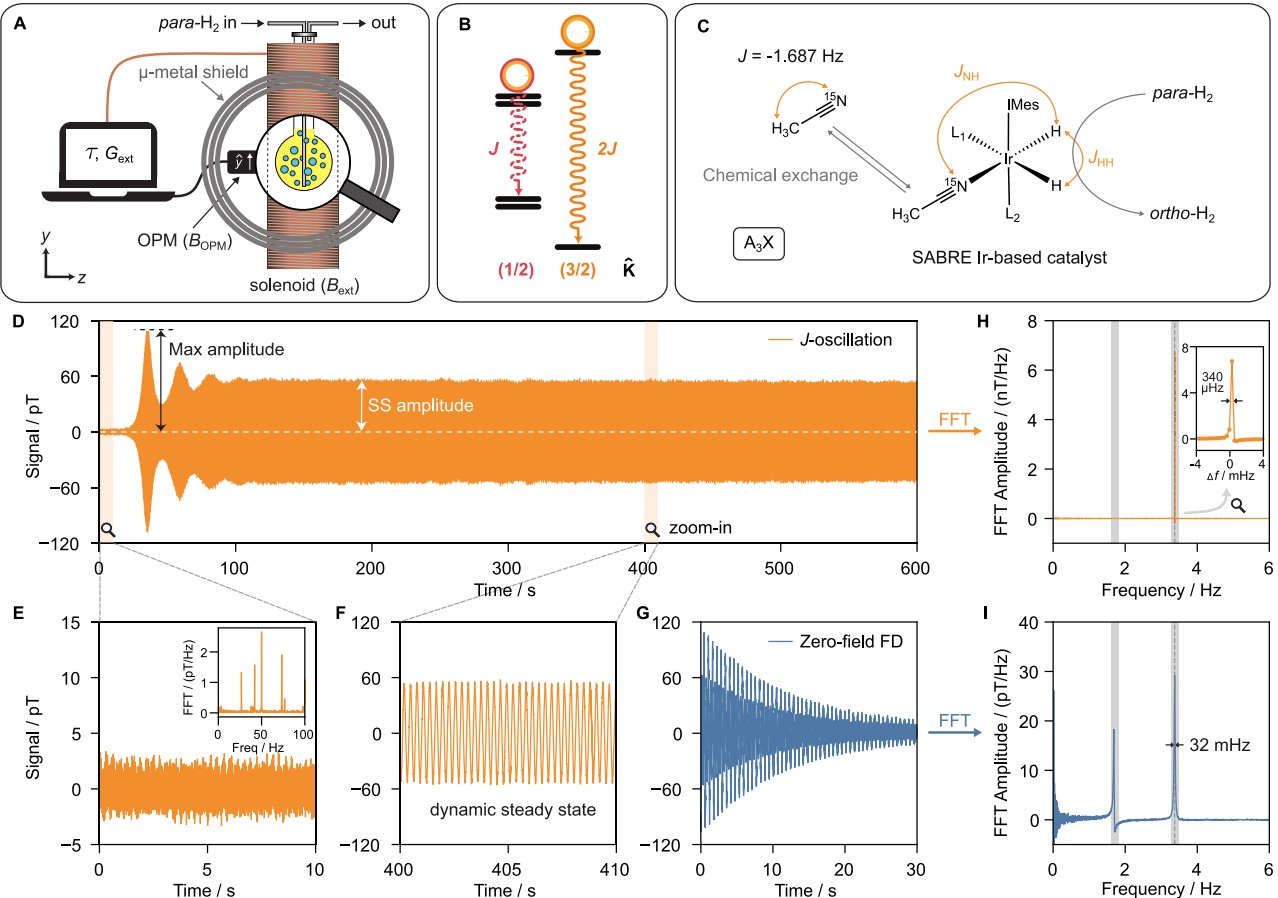

**Fig. 1 | The concept of a zero-field *J*-oscillator. A** Schematic representation of the real-time programmable feedback loop used for observing *J*-oscillators; the signal from hyperpolarized molecules is detected by an optically pumped magnetometer (OPM), processed digitally with defined feedback delay ($\tau$) and gain ($G_{ext}$), and reapplied to the molecules via a piercing solenoid. **B** Energy-level diagram of the *J*-coupling transitions of [$^{15}$N]-acetonitrile ([$^{15}$N]-ACN) at zero field. **C** [$^{15}$N]-ACN is hyperpolarized in situ at zero field via SABRE; the Ir-based catalyst facilitates spontaneous spin-order transfer from parahydrogen (*para*-H$_2$) to the population imbalances in the target molecules. **D** Experimentally recorded *J*-oscillation (time-domain) from 5% [$^{15}$N]-ACN (obtained with $\tau$ = 222 ms, $G_{ext}$ = +30), showing spontaneous emergence (<150 s) and steady-state (SS) coherent oscillation. **E, F** Zoomed-in views of **D**: **E** OPM sensor background (inset: Fast Fourier transform (FFT) showing power-line noise); **F** dynamic SS *J*-oscillation. **G** Conventional zero-field (time-domain) free decay (FD) signal from the same sample. **H, I** Spectra from the naturally abundant ACN sample measured over 3600 s (time trace in supplementary Fig. 2): **H** FFT of the *J*-oscillation over 300–3600 s shows a $\delta$-function-like peak at 2*J* with a full width at half maxima (FWHM) of 340 µHz ($\Delta f$ is referenced to 3.374 Hz); **I** Zero-field NMR signal (FFT of **G**).

frequencies depend primarily on molecular *J*-coupling constants[27], the zero-field quantum oscillators achieve significantly improved frequency stability: coherent operation of up to one hour is demonstrated (Fig. 1D). The magnetic oscillation occurs along the same axis as the feedback axis due to $\Delta m$ = 0 transitions[27], unlike the precessing magnetization with $\Delta m$ = ±1 of conventional rasers. Such an oscillator requires only a population imbalance–not a strict population inversion–achieved in situ by bubbling parahydrogen (*para*-H$_2$) into a liquid containing the activated SABRE catalyst (SABRE = signal amplification by reversible exchange), see Fig. 1C[28,29]. The catalyst enables spontaneous polarization transfer in situ at zero field, creating *J*-transition population imbalances in target molecules[30,31]. Details of the *para*-H$_2$ gas handling are given in Supplementary Methods.

## Results

### External programmable feedback loop
A central challenge in implementing zero-field quantum oscillators arises from the absence of radiation damping. In ultralow-field raser experiments–also lacking radiation damping–researchers have relied on external feedback loops[32–34]. Typically, these feedback loops detect precessing magnetization using optically pumped magnetometers (OPMs)[35], and subsequently feed signals back into the sample through

coils orthogonal to the measurement axis[36]. However, such a feedback scheme cannot be directly adopted for zero-field *J*-oscillators. This is because coherent amplification of the $\Delta m$ = 0 transitions requires that the feedback magnetic field be applied along the same measurement axis (*y*-axis,)[27].

To resolve this, we developed an external feedback loop, implemented via software control without the need for specialized hardware modifications, as schematically depicted in Fig. 1A. In our implementation, an OPM detects the *y*-cartesian component of the magnetic field generated by the sample ($B_{OPM}$). This signal is processed digitally, allowing precise control of both a tunable external feedback delay ($\tau$) and feedback gain ($G_{ext}$). The processed feedback signal is reapplied to the sample ($B_{ext}$, along the *y*-axis) using a piercing solenoid[37]. The piercing solenoid is the solenoid that goes all the way through the $\mu$-metal shield. It is designed to generate a magnetic field at the sample while producing no field at the OPM[37].

### Stability of *J*-oscillations
The *J*-oscillator was initially tested on a model system consisting of 5 % [$^{15}$N]-acetonitrile ([$^{15}$N]-ACN) dissolved in acetonitrile (ACN) solvent[29,31]. The feedback configuration enables spontaneous emergence of the quantum oscillator, as demonstrated in Fig. 1D with a feedback delay of

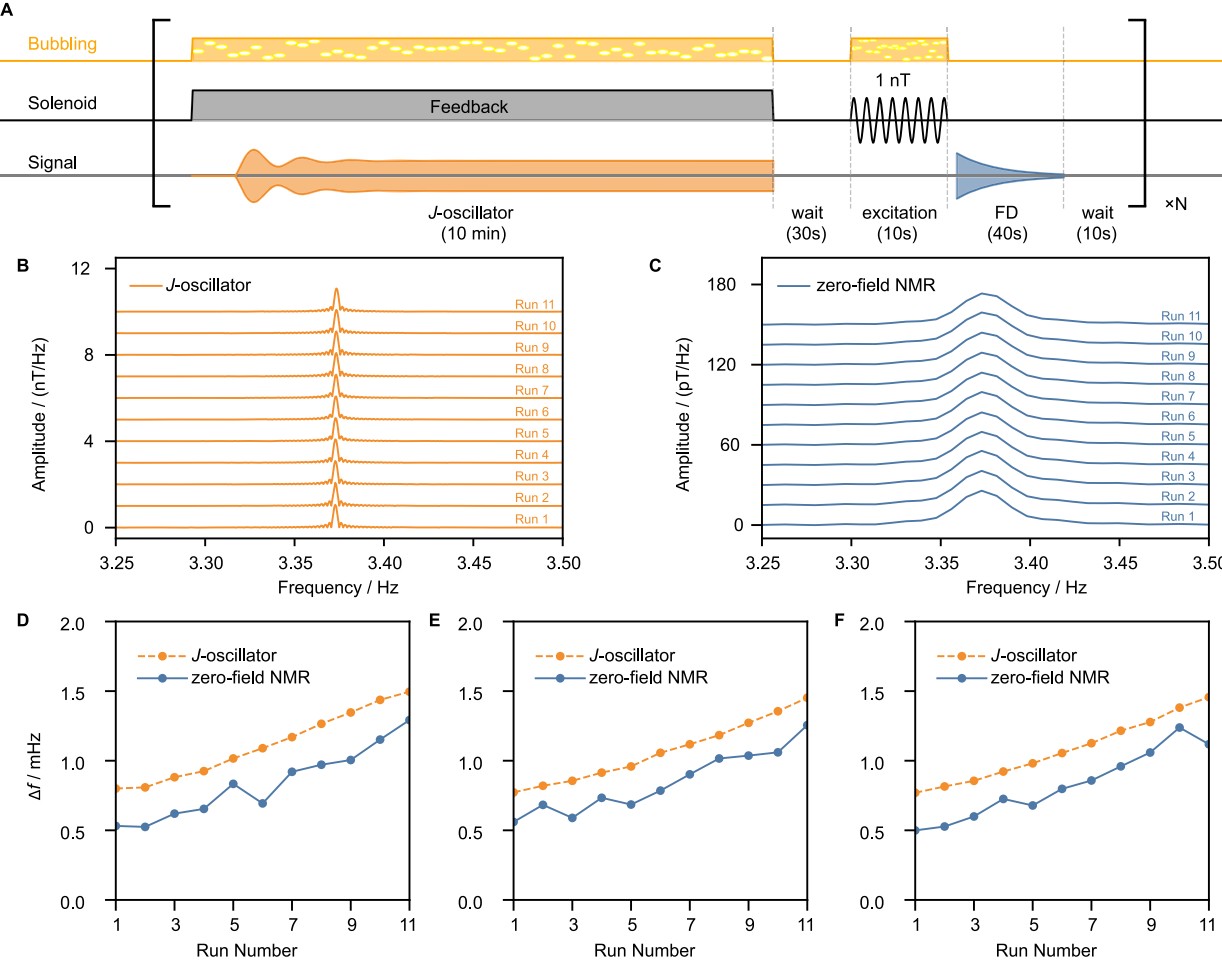

**Fig. 2 | Reproducibility of the *J*-oscillator's frequency across repeated runs over identical samples. A** Experimental sequence used to evaluate the reproducibility. Each run consists of generating *J*-oscillations ($\tau = 222$ ms, $G_{ext} = +200$), followed by a 30 s delay, 1 nT magnetic excitation at 3.374 Hz (to selectively excite the 2-*J* transition) applied during 10 s of *para*-$H_2$ bubbling, detection of the zero-field free decay (FD) signal for 40 s, and a 10 s waiting before the next run. **B** Stacked Fast Fourier transform (FFT) spectra of the steady-state *J*-oscillations (150–600 s) obtained by continuously monitoring the same sample over 11 runs; zero-filling ($\times 4$) introduces minor ripples near the peak. **C** Stacked zero-field nuclear magnetic resonance (NMR) signal obtained via FFT of the corresponding zero-field FD signals; zero-filled by a factor of 4. **D–F** Extracted frequencies of the *J*-oscillator (orange) and FD signals (blue) for three independently prepared identical samples. An example of the frequency extraction procedure is shown in Supplementary Fig. 3. A gradual increase in *J*-coupling is observed over time, while a consistent frequency offset between the *J*-oscillator and FD signals is reproducible; $\Delta f$ is referenced to 3.372 Hz.

$\tau = 222$ ms and gain $G_{ext} = +30$, requiring no pulse excitation. A zoomed-in view in Fig. 1E highlights the initial noisy background from the OPM sensor before the *J*-oscillator starts to emerge, while Fig. 1F illustrates the subsequent dynamic steady-state response. Likely, the electronic noise from the feedback loop triggers initial transitions. The signal from these spontaneous transitions is amplified by the feedback loop, and the phase is tuned for positive reinforcement. The resulting amplified field ($B_{ext}$) drives the transitions, pushing the population imbalance away from the SABRE-pumped hyperpolarized steady state[38]. With high enough external feedback gain, this SABRE-pumped population imbalance can be temporarily inverted (see Supplementary Fig. 1). Deviations from the hyperpolarized steady state manifest experimentally as transient bursts. Such bursts represent the so-called "overshooting," where the feedback-amplified transition intensities rise beyond sustainable levels[21,36]. The SABRE pumping and nuclear spin relaxation processes oppose the feedback amplification, dampening coherences and partially restoring the hyperpolarized steady state. Over successive faded bursts, the system stabilizes into what we call the "dynamic steady state" under the feedback, in which the combined effects of SABRE-pumping (replenishing population imbalances), the feedback-amplified transitions (driven by $B_{ext}$), and relaxation come into a balance.

The *J*-oscillator operating on the molecular *J*-transition was continuously recorded for 3600 s on a naturally abundant [$^{15}$N]-acetonitrile ([$^{15}$N]-ACN, 0.36 %) obtained by setting $\tau = 222$ ms and $G_{ext} = +200$. Fast Fourier transformation (FFT) (Fig. 1H) revealed a sharp, delta-function-like peak with a full-width-at-half-maximum (FWHM) of 340 μHz (Fig. 1H, inset). This linewidth is approximately the inverse of the measurement time, indicating minimal frequency drift of the experimentally observed quantum oscillation. It was later found that this linewidth has contributions of various frequencies due to a slow *J*-coupling drift (see the discussion below).

In contrast, the linewidth obtained from conventional zero-field NMR spectra does not improve with increased measurement time, as it is fundamentally limited by nuclear spin relaxation. For instance, [$^{15}$N]-ACN has a FWHM of 32 mHz for the same transition (Fig. 1H). Similarly to high-field rasers[24], the reduced linewidth achieved with the *J*-oscillators facilitates resolving closely located resonance lines.

The reproducibility of the *J*-oscillator's frequency response is confirmed in Fig. 2. We repeatedly ran the *J*-oscillator ($\tau = 222$ ms, $G_{ext} = +200$) on a naturally abundant acetonitrile sample for 10 min, then performed free-decay (FD) measurements: a 1 nT magnetic excitation at 3.374 Hz was applied during 10 s of bubbling to selectively excite the 2-*J* transition, followed by 40 s of signal detection. Figure 2B,

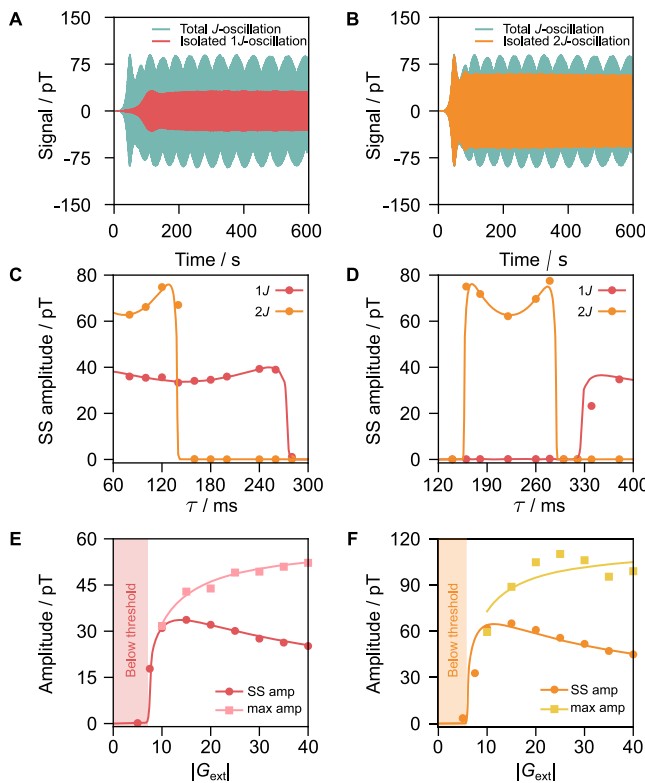

**Fig. 3 | Feedback-delay-dependent spectral selectivity and threshold dynamics of zero-field $J$-oscillators. A**, **B** $J$-oscillator on [$^{15}$N]-acetonitrile (blue) obtained with parameters $\tau = 100$ ms and $G_{ext} = -20$ demonstrates coherent generation from both 1-$J$ and 2-$J$ transitions; Fourier filtering after processing isolates 1-$J$ oscillation in (**A**) and 2-$J$ oscillation in (**B**). Panels **C**, **D** show the steady-state (SS) amplitudes of 1-$J$ and 2-$J$ oscillation versus delay ($\tau$), with a fixed feedback gain of $G_{ext} = -20$ (**C**) and $G_{ext} = +20$ (**D**), respectively. Panels **E**, **F** show the steady-state amplitude (SS amp) and maximal amplitude (max amp) dependence on the feedback gain $G_{ext}$ for oscillators operating exclusively on 1-$J$ (negative feedback, $\tau = 160$ ms) and 2-$J$ (positive feedback, $\tau = 222$ ms) transitions, respectively. Shaded areas in (**E**) ($G_{ext} < 7.2$) and (**F**) ($G_{ext} < 5.8$) indicate regions where the feedback gains fall below the threshold needed for the spontaneous oscillator emergence. For signals not reaching steady state within 600 s, the amplitude of the last 10 s was used. Solid lines in (**C**–**F**) are actual numerical simulations with the algorithm described in the "Methods" section.

**C** show the FFTs of the steady-state $J$-oscillations and the FD signals, respectively. The $J$-oscillator consistently produces narrow peaks at around the same frequencies, albeit with a small upward drift (~300 μHz per hour).

To further test the reproducibility, three independently prepared identical samples were run with the sequence eleven times per sample (Fig. 2A). Frequencies were extracted from the time-domain signals by fitting a sinusoid for the $J$-oscillator or a decaying sinusoid for the FD measurements (Supplementary Fig. 3 for an example of the fits). The results (Fig. 2D–F) show a slow, monotonic increase in the $J$-coupling constant, the exact nature of which is not fully understood. However, this effect is likely attributable to gradual changes in sample composition[39], primarily solvent evaporation during continuous bubbling (which, in turn, slowly alters the exchange dynamics with the SABRE complex). A similar drift was observed in the $J$-oscillator frequency during continuous hour-long monitoring without restarting the $J$-oscillator (see Supplementary Fig. 4, panel A). Despite this drift, the $J$-oscillator (orange traces in Fig. 2D–F) reproducibly returns to the same frequency response upon each restart, tracking the increase in $J$ with much higher precision compared to the FD measurements (blue traces in Fig. 2D-F), and remaining consistent across all three samples. A systematic millihertz-level offset in the $J$-oscillation frequency with

respect to $J$-coupling extracted from the FD measurement was observed; however, the resulting frequency offset was stable and reproducible across runs and samples. A detailed theoretical analysis of this offset will be presented elsewhere.

While both $J$-oscillators and conventional zero-field $J$-spectra demonstrate the same scaling behavior of SNR with total acquisition time (SNR $\propto \sqrt{T}$, see Supplementary Fig. 4, panel B), the error in determining transition frequency using the $J$-oscillator approach decreases more rapidly with the acquisition time due to the above-mentioned linewidth narrowing. Therefore, standard deviation in measuring precision in frequency space using $J$-oscillators approaches the Cramér-Rao lower bound[40].

## On-demand spectral editing

For a negative external feedback gain ($G_{ext} = -20$, Fig. 3C), the 1-$J$ quantum oscillator is sustained for feedback delays ranging from 60 to 275 ms (minimal delay in our system is 60 ms). Under the same negative feedback gain conditions, the 2-$J$ quantum oscillator emerges only within a narrower delay range of 60–140 ms. For positive external feedback gain ($G_{ext} = +20$, Fig. 3D), the 1-$J$ oscillator emerges spontaneously at delays ranging from 325 to 400 ms (400 ms being the maximum sampled delay in our system). Meanwhile, the 2-$J$ oscillator is sustained under delays spanning from 160 ms to 290 ms. Thus, the emergence of 1-$J$ and 2-$J$ quantum oscillators exhibits different dependencies on the externally applied feedback delays.

This behavior is due to the frequency-dependent phase shifts introduced by the delays. To gain more insight, the delay ($\tau$) is converted to the corresponding feedback phase lag ($\varphi$) at the operating $J$-transition frequency $f$, calculated as $\varphi = 2\pi f \tau$ as presented in Supplementary Table 1. Combining the results from both plots, it was found that the 1-$J$ quantum oscillator is sustained across the range of phase lags from around 3.4 to 6.1 radians. Similarly, the 2-$J$ oscillator remains active within a comparable phase lag interval, between approx. 3.4 and 6.1 radians. These phase intervals are symmetric around $3\pi/2$, spanning roughly $\pm 0.86 \cdot (\pi/2)$ around this central value.

The dependence of the $J$-oscillator operation on phase can be understood by decomposing the feedback field mathematically into two components relative to the field generated by the sample. The first component aligns with the signal ("in-phase," phase lag 0) or is opposite to it ("anti-phase," phase lag $\pi$). The second component, the "quadrature" component, is shifted by a quarter cycle (phase lag $\pm\pi/2$) relative to the signal. Only this quadrature component effectively contributes to amplifying the selected $J$-transitions. To quantify this amplifying contribution, we define an "effective external feedback gain," represented as $G_{ext} \sin \varphi$[41]. For sustained quantum oscillations to emerge, the absolute value of this effective gain must exceed a specific threshold (discussed below). Moreover, the sign of the effective gain must match the direction of the hyperpolarized population imbalances: in the current case, a negative effective gain is required, corresponding to a population inversion between the selected $J$-transitions. Consequently, the dependence of quantum oscillator emergence on the externally imposed phase lag provides a convenient method to selectively amplify individual $J$-transitions at specific frequencies, demonstrating the capability for on-demand spectral editing (see Fig. 4).

## Threshold dynamics

Figure 3E, F show the dependence of the steady-state and the initial burst amplitudes on the external feedback gain ($G_{ext}$) for quantum oscillators operating selectively at the 1-$J$ (Fig. 3E, $\tau = 160$ ms) and 2-$J$ (Fig. 3F, $\tau = 222$ ms) transitions, respectively. These delays ensure that the feedback corresponds to a phase lag near $3\pi/2$ for each peak, resulting in a maximal contribution of the quadrature component ($|\sin \varphi| = 1$). In both plots, the steady-state amplitudes initially increase with feedback gain but decline at higher gains, mirroring the

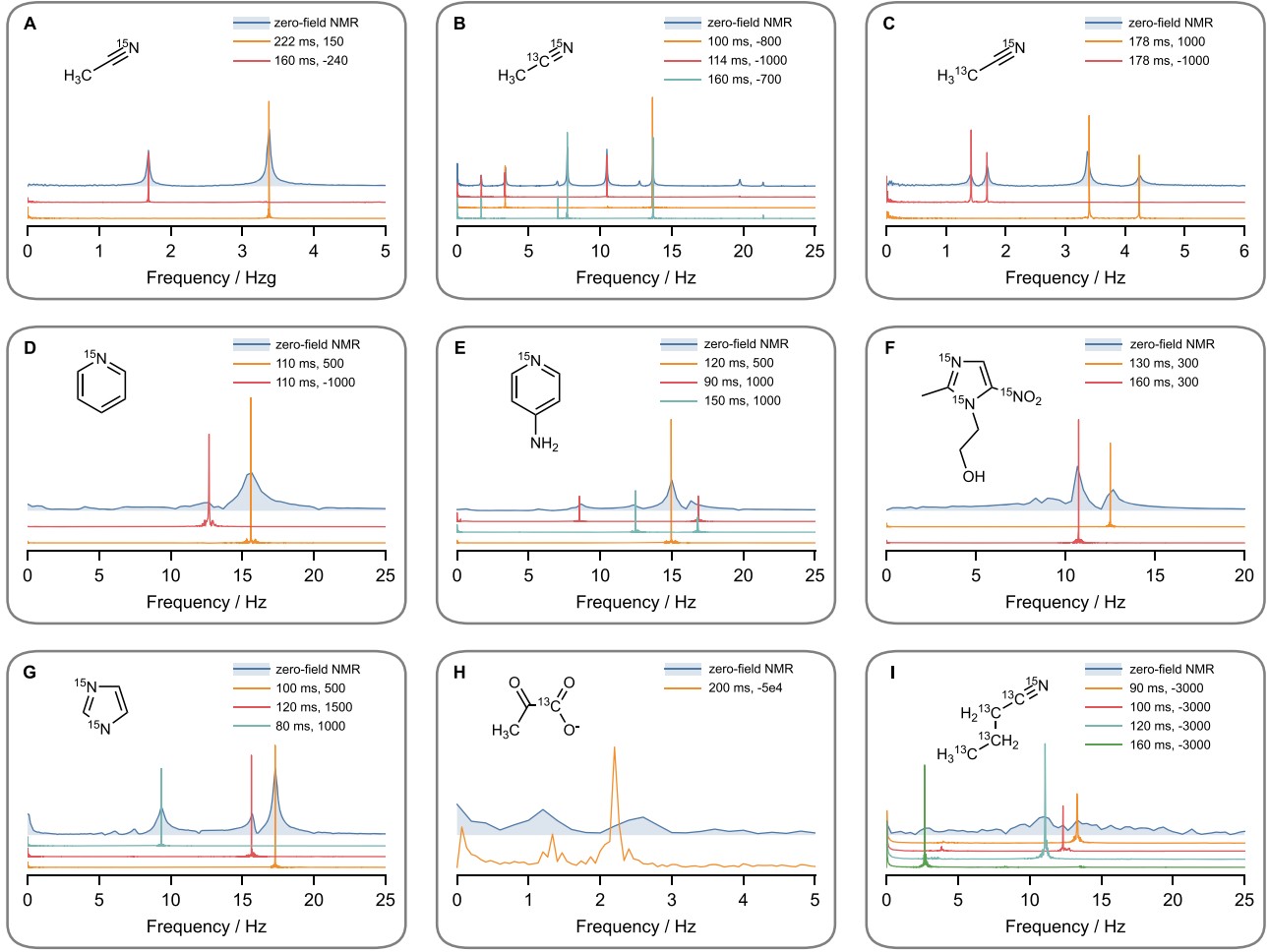

**Fig. 4 | Quantum *J*-oscillators realized on various molecules.** Comparison between conventional zero-field nuclear magnetic resonance (NMR) spectra (blue shaded traces) and quantum *J*-oscillator signals from exemplary chemical systems: **A** naturally abundant acetonitrile (0.36% [$^{15}$N]-ACN, 99.64% [$^{14}$N]-ACN); **B** 1% [2-$^{13}$C,$^{15}$N]-ACN in [$^{14}$N]-ACN; **C** 1% [1-$^{13}$C,$^{15}$N]-ACN in [$^{14}$N]-ACN; **D** [$^{15}$N]-pyridine; **E** 4-amino[$^{15}$N]-pyridine; **F** [$^{15}$N$_3$]-metronidazole; **G** [$^{15}$N$_2$]-imidazole; **H** [1-$^{13}$C]-pyruvate; **I** [U-$^{13}$C, $^{15}$N]-butyronitrile. Each *J*-oscillator is characterized by two parameters indicated in the figure legends: the feedback delay ($\tau$) followed by the external gain ($G_{ext}$).

conventional maser/laser dynamics where output peaks at an optimized resonator quality factor[42]. The fact that higher gains result in lower SS amplitudes is also indicated by the minima observed in Fig. 3C (1-*J* at 160 ms) and 3D (2-*J* at 220 ms). These minima correspond to conditions with the feedback phase lag of approximately $3\pi/2$, where the effective feedback gain has the biggest amplitude. At higher gains, the maximal amplitude of the first burst plateaus as the hyperpolarization-driven population imbalance is converted into coherence.

The observed *J*-oscillations in Fig. 3E, F are affected by two contributions: the passive magnetic intrinsic gain of the system $G_{int}$, and the actively applied external feedback gain $G_{ext}$. The intrinsic gain $G_{int}$, which is analogous to the conventional gain definition in laser physics[9], quantifies the ability of the system to amplify magnetic fields. Specifically, the intrinsic gain $G_{int}$ is defined as the ratio between the amplitude of the magnetic field produced by the sample (as measured by the OPM) and the amplitude of the externally applied AC field that acts on the sample (see "Methods" section for details). For sustained quantum oscillations to emerge spontaneously, the total gain from combined internal and external contributions must exceed unity:

$$|G_{ext} \cdot G_{int}| > 1. \tag{1}$$

For the cases shown in Fig. 3E, F, the threshold external gains $G_{ext}^{th}$ required to initiate sustained oscillations, extracted from the numerical simulation illustrated in figures, are approximately 7.2 for the 1-*J* transition and 5.8 for the 2-*J* transition, respectively, as highlighted by the shaded areas in the respective figures. Additionally, simulations of the exact same modeled spin systems yield intrinsic gain values $G_{int}$ of approximately 0.138 and 0.172 for the 1-*J* and 2-*J* transitions, respectively (see Supplementary Fig. 5 and "Methods" section for detailed numerical procedures). Overall, these extracted values meet the theoretical threshold condition defined by Eq. (1).

By tuning the external feedback gain, it is possible to achieve the *J*-oscillator behavior under challenging conditions such as low molar polarization (product of molecular concentration and nuclear polarization) or rapid relaxation, where intrinsic gain alone would be insufficient to initiate the emergence of the oscillator[18]. Increasing the external feedback gain ($G_{ext}$) effectively lowers the required polarization threshold, allowing the observation of the quantum oscillator behavior for various molecules, see the section below. The minimum SNR required for self-oscillations is therefore set by the maximum practically applicable $G_{ext}$ (see the "Methods" section), which is limited by solenoid-sensor cross-talk: at large feedback, leakage noise is fed back into the loop, causing instability. This solenoid leakage originates mainly from imperfect coil winding, and improved coil construction

can mitigate the problem. Conversely, when the external feedback gain is limited, only the *J*-transitions with sufficiently large intrinsic gain can surpass the threshold condition. Exploiting this property enables selective excitation of individual resonance lines within spectra containing densely overlapping peaks (see Fig. 4).

## Generality across various molecules

Figure 4 presents spectra of *J*-oscillators alongside their corresponding zero-field NMR spectra for various illustrative chemical systems. Unless otherwise noted, the *J*-oscillators were acquired for 10 min (except for the trace in Fig. 4D collected with $G_{ext} = -1000$ and Fig. 4H, I). The corresponding FFT spectra were normalized using division by the total number of data points (Convention I, see Supplementary Notes 2), allowing for the direct comparison between quantum oscillator-based and zero-field NMR spectra, despite differences in acquisition times. The sample preparation is discussed in the "Methods" section.

Acetonitrile (ACN) is revisited first. Figure 3A shows *J*-oscillations generated by naturally abundant acetonitrile ([$^{15}$N]-ACN, 0.36%). Despite roughly ten-fold isotopic dilution compared to the previously discussed samples (5%), we successfully achieved sustained oscillations by compensating the reduced intrinsic gains with increased external feedback gain. Additionally, isotopically labeled ACN molecules with 1% abundance of $^{15}$N were tested: specifically, [1-$^{13}$C,$^{15}$N]-ACN and [2-$^{13}$C,$^{15}$N]-ACN (Fig. 4B, C).

The approach also works for more complex spin systems. A longer-chain nitrile, [U-$^{13}$C,$^{15}$N]-butyronitrile (Fig. 4I) and several heterocycles−[$^{15}$N]-pyridine, 4-amino[$^{15}$N]-pyridine, [$^{15}$N$_3$]-metronidazole, and [$^{15}$N$_2$]-imidazole (Fig. 4D–G)−were tested. They all show complicated zero-field spectra, yet individual lines could still be "picked out" and driven into clean, millihertz-wide *J*-oscillations by optimizing the feedback parameters. The *J*-oscillators operating on [$^{15}$N]-pyridine under a higher feedback gain ($G_{ext} = -3000$) are given in Supplementary Fig. 6. Under specific feedback conditions, near-DC *J*-oscillations were observed in butyronitrile, and sub-0.5 Hz *J*-oscillations were seen in metronidazole and 4-amino-pyridine (Supplementary Figs. 7, 8). Whether they represent extremely low-frequency *J*-transitions or some nuclear-paramagnetic effect warrants further investigations.

Finally, we established *J*-oscillators in the biologically relevant probe [1-$^{13}$C]-pyruvate (Fig. 4H). Due to the poor hyperpolarization at elevated temperature conditions (heat up by the OPM), together with the broad Hz-scale FWHM, the conventional zero-field NMR spectrum of this molecule (blue trace) exhibited poor SNR (~4). But the resulting *J*-oscillator (orange trace) showed significantly improved SNR (~70 for 2-*J*) and a narrower linewidth. A frequency shift of both transitions was detected because the feedback loop coupled the two transitions and "drew" the peaks toward each other[41]. One can envision *J*-oscillator experiments with cell cultures aimed at monitoring continuous pyruvate-to-lactate production−something hard to achieve by conventional zero-field NMR.

To illustrate the analytical power of the *J*-oscillator method, we investigated a series of mixtures that all contain 50 mM pyridine, 50 mM 4-aminopyridine and 3 mM of the Ir-based catalyst, but differed in the $^{14}$N/$^{15}$N isotopic ratio of the two substrates. Conventional zero-field NMR spectra of these samples show that both molecules have their main peaks overlapping at around 15 Hz (Supplementary Fig. 9). In contrast, the *J*-oscillator approach yields well-resolved peaks for each molecule. Figure 5 shows the resulting spectra, all obtained using an identical feedback delay ($\tau = 115$ ms) but varying feedback gain.

For the mixture in which both substrates were 100% $^{15}$N-labeled, two narrow resonances with approximately equal intensities emerge at moderate feedback gains. Comparison with the entire set of data reveals that the lower-frequency peak originates from 4-amino-[$^{15}$N]-pyridine ([$^{15}$N]-FamPy), whereas the higher-frequency line stems from

[$^{15}$N]-pyridine ([$^{15}$N]-Py). With high enough feedback gain, the *J*-oscillations from molecules that are 50% $^{15}$N-labeled can also be obvserved (Supplementary Fig. 10). In the sample that contained 100% [$^{15}$N]-Py and 50% [$^{15}$N]-FamPy, the right peak is about twice as strong as the left one, giving the expected 2:1 ratio that reflects upon its isotopic composition. The opposite 1:2 intensity ratio is observed when the enrichment levels are swapped (see Supplementary Fig. 10). Furthermore, as illustrated in Supplementary Fig. 11, adjusting the feedback delay allows the *J*-oscillation peaks of individual molecules to be selectively excited and distinguished within the same sample.

The experiments also show several interesting dynamical effects. First, whenever the feedback gain is tuned such that only one molecule oscillates, its resonance drifts to higher frequencies as the gain is increased. Second, when both oscillators are excited, resonances from the two molecules push each other apart, and the frequency separation increases with gain. These shifts do not reflect changes in the intrinsic *J*-couplings; they arise because the feedback field loop couples the spin dynamics of the two molecules. A detailed quantitative treatment is beyond the scope of the present work and will be reported elsewhere. Note that the shifts are reproducible and can be calibrated through careful tuning of the feedback parameters so that the intrinsic *J*-couplings can be extracted.

The results suggest that the *J*-oscillator method can assist in analyzing complex mixtures containing structurally similar molecules, particularly in situations where conventional zero-field NMR does not provide sufficient resolution to distinguish signals[24]. Since the method relies on SABRE hyperpolarization, it is only applicable to molecules with sufficient affinity to the SABRE catalyst, thereby restricting the range of detectable species. Nonetheless, this limitation can be mitigated in the future by employing alternative hyperpolarization schemes such as SABRE-relay, Overhauser-DNP, or even thermally polarized samples under continuous flow, to broaden the scope of molecules accessible to this method.

## Toward many-body non-linear spin dynamics

As the applied feedback gain is further increased, a rich variety of nonlinear dynamics−including the emergence of multiple lines and chaos−arises from the imposed feedback that couples different *J*-transitions. These phenomena highlight the unique potential of zero-field *J*-oscillators as convenient platforms for systematic exploration of nonlinear spin dynamics.

A key advantage of the zero-field *J*-oscillators (compared to Zeeman Rb-Xe co-magnetometers[36] or high-field rasers)[18] is their potential to operate as engineered multi-mode systems. Such behavior can be achieved by applying a static bias field, which is known to split a single *J*-transition into multiple transitions[43]. Existing theoretical work predicts that the multi-mode systems of this type, when subject to active feedback, could undergo transitions in dynamical behavior ranging from mode collapse to frequency combs formation and chaos, depending on the separation between the transitions[44]. This mechanism suggests a potential route for future studies of nonlinear spin dynamics using zero-field *J*-oscillators. In contrast, high-field rasers oscillate at the nuclear Larmor frequency, and additional modes arise from intrinsic *J*-coupled multiplets, these spacings are set by molecular parameters and are therefore not readily tunable.

Digital signal processing provides an opportunity for future extensions of the feedback scheme. Advanced digital filtering algorithms could mitigate experimental imperfections; for example, OPM temperature drift could be suppressed by incorporating a DC-blocking filter. In addition, more complicated feedback protocols based on derivatives or nonlinear functions of the detected signal may provide access to complex dynamical regimes, such as chaos or time-crystalline-like behavior in liquid-state molecular systems[45]. These possibilities remain to be explored experimentally in future work.

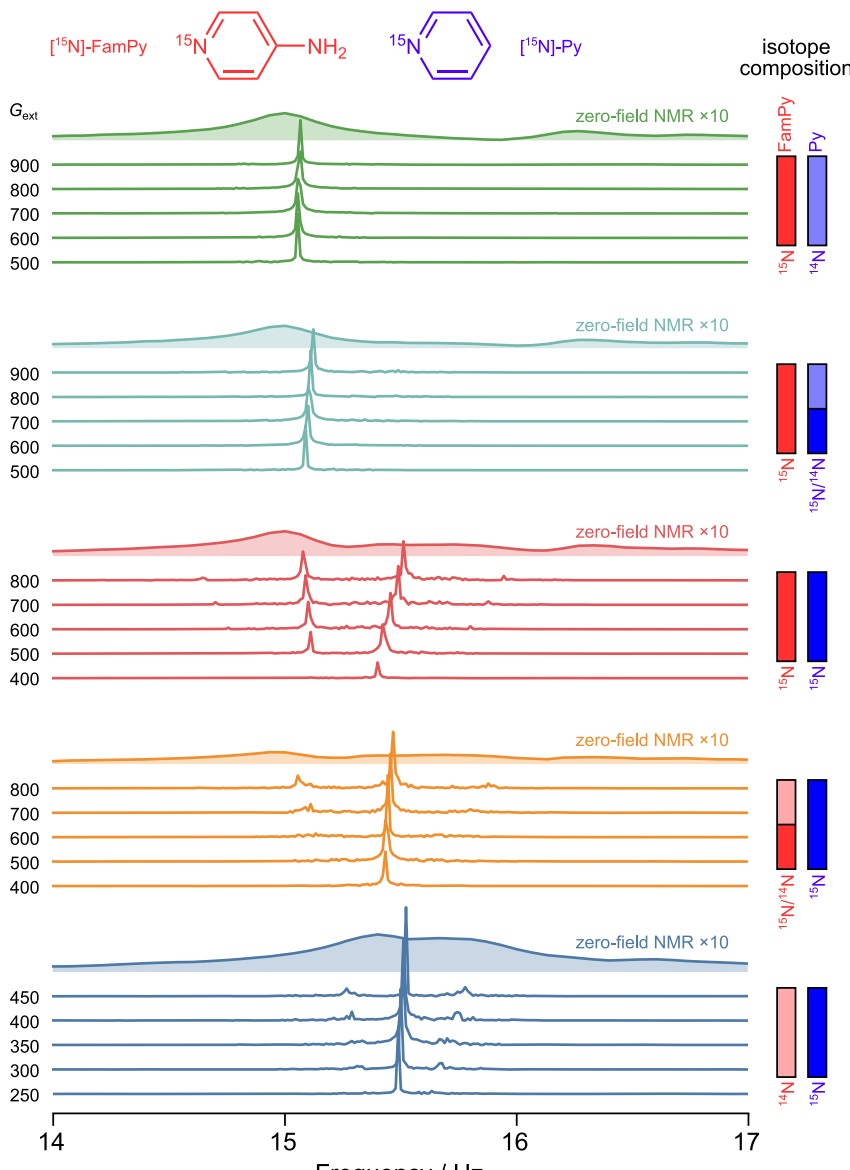

**Fig. 5 | *J*-oscillators operated on pyridine/4-aminopyridine mixture with different ¹⁴N/¹⁵N isotopic compositions.** Each sample contains 50 mM pyridine, 50 mM 4-aminopyridine, and 3 mM of the Ir-based catalyst in methanol, with only the ¹⁵N isotopic enrichment of the two substrates varied. For each sample, the amount of substrate is indicated by a pair of vertical bars: the left red bar for 4-aminopyridine (FamPy) and the right blue bar for pyridine (Py). The shading of each bar denotes the degree of ¹⁵N enrichment: solid (100% ¹⁵N enriched), light (natural-abundance ¹⁴N), and half solid/half light (50% ¹⁵N enrichment). The traces measured on the same sample are plotted with the same color, with the feedback delay fixed at $\tau = 115$ ms and the applied external feedback gain annotated to the left of each trace. The Fourier spectra are obtained by analyzing the final 90 s of the recorded 180 s of *J*-oscillation data in order to report on the steady state and avoid the burst/transient. For reference, the conventional zero-field nuclear magnetic resonance (NMR) spectra of each sample are shown as shaded traces, scaled up by a factor of 10 for clarity.

## Methods

### External feedback loop apparatus

The hardware components of the feedback loop system are as follows. The signal produced by the sample is measured using a commercially available optically pumped magnetometer (model QuSpin Gen-2) with a sensitivity of around 15 fT/$\sqrt{\text{Hz}}$ in a 3–100 Hz band and a calibration factor of 1000/2.7 pT/V. The analog output from the OPM is digitized with a National Instruments (NI) analog input card (model NI 9239) at a sampling rate of 2000 Hz. This digitized signal is transmitted to the computer via the NI Measurement & Automation Explorer interface. A Python-based real-time processing algorithm running on the computer set the delay and gain parameters to the acquired signal. The processed signal is then converted back to an analog form using an analog output (model NI 9263) at a sampling rate of 2000 Hz as well and delivered to a piercing solenoid, which generates the feedback magnetic field.

### Sample preparation

Samples were prepared in a nitrogen atmosphere and transferred into a spherical NMR tube (outer diameter 10.5 mm, inner diameter 8.5 mm)[31] that can be adapted to the *para*-H₂ gas-handling setup. Samples were prepared of a common Ir-precursor [Ir(IMes)(COD)Cl] dissolved in acetonitrile or methanol, c.f. detailed sample composition is provided in Supplementary Table 2. The following substrates and

solvents were purchased from Sigma Aldrich and used without further purification: (A) acetonitrile (0.36% [$^{15}$N]-ACN and 99.64% [$^{14}$N]-ACN), (B) 1% [2-$^{13}$C,$^{15}$N]-ACN in 99% [$^{14}$N]-ACN, and (C) 1% [1-$^{13}$C,$^{15}$N]-ACN in 99% [$^{14}$N]-ACN; (D) [$^{15}$N]-pyridine, (E) 4-amino[$^{15}$N]-pyridine, (F) [$^{15}$N$_3$]-metronidazole, (G) [$^{15}$N$_2$]-imidazole, (H) [1-$^{13}$C]-pyruvate, and (I) [U-$^{13}$C, $^{15}$N]-butyronitrile, see Figure 4. An additional co-substrate[29,46,47] was introduced to stabilize hyperpolarization transfer efficiency of samples (A, B, C, H, and I), which are listed in Supplementary Table 2.

### Programmable feedback algorithm

**Variable definitions and constraints.** Apart from $G_{ext}$ and $\tau$, the following variables are defined for the implementation of the feedback loop:

- $R_{buf}$, $W_{buf}$ (Size $N_b$): Cyclic buffers storing the history of input (read) and output (written) samples, respectively; both are initialized with zero values.
- $R_{in}$, $W_{out}$ (Size $N_c$): Arrays for the current voltage samples read from NI 9239 (AI) and to be written to NI 9263 (AO); both are initialized with zero values.
- $D$: The set feedback delay in samples.
- $D_p = \tau_{pd} \times f_s$: The hardware propagation delay in samples.
- $N_i$: The number of zero-valued initial samples sent to the AO at the start of the experiment.
- $i$: Index pointer for the current reading and writing position.

To ensure smooth operation of the loop and to prevent exceeding the available range of the cyclic buffers, the buffer and memory sizes must satisfy the conditions $D \geq N_i \geq N_c$ and $N_b \geq \max(N_c + (D - N_i), N_0 + d)$. In practice, we typically set $N_b = 4D$ and ensure $N_i \geq N_c + 20f_s$ (where $f_s$ is in kHz) to avoid potential hardware overheads.

**The active feedback protocol.** The feedback loop runs continuously, with the analog output (AO) and analog input (AI) modules sharing the same clock and start trigger. Since the loop requires memory of previously written samples to generate delayed outputs, we begin by writing $N_i$ zero-valued samples to the AO module. Given that the samples produce zero signal initially, we assume that all prior reads and writes are also zero, meaning the buffers $W_{buf}$ and $R_{buf}$ are initialized with zero values. During operation, while the AO module is still outputting its current buffer, the AI module simultaneously acquires new $N_c$ samples from the OPM output. These samples are processed according to Algorithm 1 and written into the AO memory buffer before the previous data in AO memory are fully run out. This continuous overlap between acquisition, processing, and output ensures that the AO buffer never becomes empty, allowing uninterrupted feedback loop execution.

The programmable external active feedback algorithm is presented in Algorithm 1. It is essential to feed back only the sample-generated signals back to the sample, excluding the feedback field produced by the solenoid itself or amplified OPM noise. Although the piercing solenoid is designed to achieve this by producing a magnetic field at the sample location while generating no measurable field at the OPM. In practice, the solenoid is not perfectly isolated, and a small portion of its field leaks outside. We mitigate the effect of this leakage by a digital compensation step (Step 2 in Algorithm 1) which subtracts the expected leakage from the OPM readout based on the previously written output and a calibrated leakage factor $C_l$. The solenoid leakage factor $C_l$ is defined as the ratio between the voltage measured by the OPM due to solenoid field leakage (via the NI 9263) and the input voltage applied to the solenoid (via the NI 9239). The programmable feedback delay is subsequently implemented by retrieving the processed signal from the cyclic buffer $R_{buf}$ at an index offset by $D$ samples relative to the current write position (Step 5 in Algorithm 1).

**Algorithm 1.**

> **Input:** Real-time voltage data stream from AI
> **Output:** Real-time output voltage stream to AO
>
> ▷ $R_{buf}$ and $W_{buf}$ are cyclic $(\mathrm{mod}\ N_b)$.
1  **while** true **do**
> ▷ 1) Acquire
2  $R_{in} \leftarrow$ Read $N_c$ samples from NI 9239;
> ▷ 2) Leak cancellation + unit conversion
3  **for** $k \leftarrow 0$ **to** $N_c - 1$ **do**
4  $\quad j \leftarrow (i + k - D_p)\ \mathrm{mod}\ N_b$;
5  $\quad R_{in}[k] \leftarrow R_{in}[k] - C_l \cdot W_{buf}[j]$;
6  $\quad R_{in}[k] \leftarrow R_{in}[k] \cdot \frac{1000}{2.7}$;
> ▷ 3) Record and Filter
7  Save $R_{in}$ to file;
8  Apply moving-average to $R_{in}$
> ▷ 4) Update $R_{buf}$
9  **for** $k \leftarrow 0$ **to** $N_c - 1$ **do**
10  $\quad R_{buf}[(i + k)\ \mathrm{mod}\ N_b] \leftarrow R_{in}[k]$;
> ▷ 5) Apply feedback delay and gain
11  **for** $k \leftarrow 0$ **to** $N_c - 1$ **do**
12  $\quad W_{out}[k] \leftarrow R_{buf}[(i + N_i - D + k)\ \mathrm{mod}\ N_b]$;
13  $\quad W_{out}[k] \leftarrow W_{out}[k] \cdot G_{ext} \cdot \frac{R_0}{C_0} \cdot 10^{-6}$;
> ▷ 6) Output
14  Output $W_{out}$ to NI 9263;
> ▷ 7) Update $W_{buf}$
15  **for** $k \leftarrow 0$ **to** $N_c - 1$ **do**
16  $\quad W_{buf}[(i + N_i + k)\ \mathrm{mod}\ N_b] \leftarrow W_{out}[k]$;
> ▷ 8) Advance the loop index
17  $i \leftarrow i + N_c$;

**Calibration.** Due to the frequency-dependent response of the OPM, leakage is calibrated at 5 Hz (proximal to our < 20 Hz region of interest), while high-frequency leakage (> 4 Hz) is suppressed via moving-average filters (Step 3 in Algorithm 1). Standard measurements employ a 10 ms window; pyruvate data utilized a 40 ms window applied sequentially three times. The corresponding filter amplitude responses are shown in Supplementary Fig. 12. With active feedback, the OPM sensitivity obtained under different moving-average settings is presented in Supplementary Figs. 13–14. The OPM noise floor remains constant across increasing gains, confirming that the $J$-oscillator sensitivity is not limited by amplifier noise.

Calibration of the leakage factor $C_l$ and propagation delay $\tau_{pd}$ utilizes a synchronized 5 Hz sine test with shared AO/AI clocks and start trigger. A test signal drives the solenoid (from NI 9263, $R_0 = 430$ k$\Omega$) while the OPM response is recorded (by NI 9239). The propagation delay $\tau_{pd}$ is determined from the signal onset lag, and $C_l$ is derived from the output-to-input amplitude ratio.

In our case, driving the solenoid with 10.9 nT amplitude sinusoid at 5 Hz produced 8.6 pT, 5 Hz sinusoid at the OPM, giving $C_l \simeq 8.6$ pT/ 10.9 nT $\simeq 7.9 \times 10^{-4}$. The compensation algorithm gives an additional ~200× suppression (verified via independent Helmholtz coil injection and residual measurement). The maximal stable external feedback gain is bounded by the inverse compensated leakage, $|G_{ext,max}| \lesssim 1/ (C_l \times 200) \simeq 2.5 \times 10^5$. In practice, $|G_{ext}|$ is restricted to < 20% of this limit to prevent leakage-induced instability.

To achieve the intended feedback delay $\tau$, the set delay in the program must be adjusted to account for the intrinsic hardware propagation delay and the additional delay introduced by the digital filter. As a result, the set delay is $D = \tau \times f_s - D_p - D_f$, where $D_p$ is the hardware propagation delay and $D_f$ is the delay caused by the filter. Standard

experiments used a 10 ms window (order 1, $D_f = 5$ ms), while pyruvate measurements required stronger smoothing (60 ms window, order 3). For example, measuring ACN (2-$J$ transition) with target $\tau = 222$ ms, hardware delay $D_p = 12$ ms, and filter delay $D_f = 5$ ms, the set delay is $D = (222 - 12 - 5) \times f_s$. We also choose $N_i = 160 f_s$, $N_c = 100 f_s$, and $N_b = 4D$ to satisfy the buffer requirements.

## Theory of $J$-oscillators

The dynamics of the $J$-oscillator signal can be simulated using a master equation:

$$\frac{d}{dt}\widehat{\rho}(t) = -i[\widehat{H}_0 + \widehat{V}(t), \widehat{\rho}(t)] + \widehat{\widehat{R}}\widehat{\rho}(t) + \widehat{P}, \qquad (2)$$

where $\widehat{\rho}(t)$ is the density operator of the molecule, $\widehat{H}_0$ represents the $J$-coupling Hamiltonian, $\widehat{V}(t)$ represent the coupling of the molecule with the delayed feedback filed, $\widehat{\widehat{R}}$ accounts for relaxation processes, including intramolecular dipolar interactions and intermolecular paramagnetic effects modeled via fluctuating random fields[48,49] and and $\widehat{P}$ represents SABRE pumping.

Specifically, the feedback field couples to the sample magnetization along the detection axis,

$$\widehat{V}(t) = -G_{\text{ext}} \cdot B_{\text{OPM}}(t - \tau) \cdot \sum_i \gamma_i \widehat{I}_{i,y}, \qquad (3)$$

where $B_{\text{OPM}}(t)$ represent the field generated by the sample as measured by the OPM. The operator $\widehat{I}_{i,y}$ denotes the angular momentum operator of the $i$-th nuclear spin along the detection axis, and $\gamma_i$ is its corresponding gyromagnetic ratio.

The forms of the Hamiltonians, relaxation superoperators, and pumping operator are provided in the Supplementary Notes 1. The master equation was solved numerically using a second-order Strang splitting scheme[50], which separates coherent evolution from dissipative dynamics. Full implementation details are given in the Supplementary Notes 1.

## The intrinsic magnetic gain of samples

To quantify the intrinsic magnetic gain of the sample, $G_{\text{int}}$, we add the interaction between the sample and a weak magnetic field applied via the piercing solenoid. The applied field $H(t)$ (magnetic field intensity) oscillates at frequency $f$, the system dynamics is determined by,

$$\frac{d}{dt}\widehat{\rho}(t) = -i[\widehat{H}_0 + \widehat{V}(t), \widehat{\rho}(t)] + \widehat{\widehat{R}}\widehat{\rho}(t) + \widehat{P}, \qquad (4)$$

where the perturbation Hamiltonian (for [$^{15}$N]-acetonitrile) is,

$$\widehat{V}(t) = -\mu_0 H(t) \cdot (\gamma_{^{15}N}\widehat{S}_y + \gamma_{^1H}\widehat{K}_y). \qquad (5)$$

In the dynamic steady state, the magnetization $M(t)$ of the sample oscillates at the same frequency $f$ as the applied field $H(t)$. This magnetization generates a detectable magnetic flux at the OPM. When the applied field amplitude is sufficiently small (i.e., when the perturbation from the applied field is weak compared to the system's relaxation rate), the ratio between the amplitudes of the measured magnetic flux (at the sensor) and the applied magnetic flux becomes independent of the applied field amplitude. Instead, it depends solely on the frequency $f$.

We define the internal magnetic gain $G_{\text{int}}(f)$ as this frequency-dependent response,

$$G_{\text{int}}(f) = \frac{r^3}{3d^3} \cdot \frac{|M(t)|}{|H(t)|} = \frac{r^3}{3d^3}|\chi(f)|, \qquad (6)$$

where $r$ is the radius of the sample and $d$ is the distance between the sensor and sample, $|\chi(f)|$ is a frequency-dependent complex magnetic susceptibility.

Simulation shows maximal internal gains at $f = J$ and $f = 2J$, with internal gains $G_{\text{int}}(J) \approx 13.8\%$ and $G_{\text{int}}(2J) \approx 17.2\%$, respectively (Supplementary Fig. 5). As discussed, the threshold external feedback gain ($G_{\text{ext}}^{\text{th}}$) required for maser oscillations to spontaneously emerge corresponds to the inverse of the on-resonance internal gains, $G_{\text{ext}}^{\text{th}} = 1/G_{\text{int}}(f_0)$. For 1-$J$ and 2-$J$ transitions, this yields $G_{\text{ext}}^{\text{th}} \approx 7.2$ and $G_{\text{ext}}^{\text{th}} \approx 5.8$, respectively, consistent with the results in Fig. 3E, F.

## Data availability

Source data are available in the *figshare* repository at https://doi.org/10.6084/m9.figshare.30992242.

## Code availability

Custom code used for the active feedback algorithm is provided in the Supplementary Software.

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

## Acknowledgments

This work was supported by the Alexander von Humboldt Foundation in the framework of the Sofja Kovalevskaja Award and in part by the Deutsche Forschungsgemeinschaft (DFG, German Research Foundation) in the framework of the collaborative research center "Defects and Defect Engineering in Soft Matter" (SFB1552) under Project No. 465145163 and under the DFG/ANR grant BU 3035/24-1. The publication is funded by the Open Access Publishing Fund of GSI Helmholtzzentrum für Schwerionenforschung.

## Author contributions

Conceptualization: J.X., R.K., O.T., D.B., and D.A.B. Supervision: D.A.B. Writing - original draft: J.X. Writing - review & editing: J.X., R.K., O.T., D.B., and D.A.B.

## Funding

## Competing interests

The authors declare no competing interests.
