## [Transparent Peer Review file · Nature Communications]

Quantum Magnetic J -Oscillators

Corresponding Author: Professor Danila Barskiy

Version 0:

Reviewer comments:

Reviewer #1

(Remarks to the Author)

See attached review in PDF format. I expressed some criticism and suggested changes to improve the manuscript. Overall, I recommend accepting the manuscript for publication after minor revisions.

Reviewer #2

(Remarks to the Author)

The authors report a novel and useful approach to generating a steady state signal that is insensitive to temporal variations in the Zeeman interaction. They base their work on driving the scalar coupling in zero magnetic field.

The work is well described and could be published as is, however there are a few points they may wish to consider.

There are a few places where a steady state response is somewhat confused with a coherence. An example is where a linewidth is reported for a steady state response. Since this is steady state, the response will appear for all time (provided that the spin polarizer continues to function). I would change the language and make clear that the reported linewidth is artificial and is not an indication of a sensor sensitivity. The same factor makes the use of the method for time-crystals somewhat less interesting.

The use of an amplifier in the feedback loop to drive the response is interesting, though it has been done before including by Laukien (to reduce radiation damping). Is there any analysis on the effects of noise from the addition of the amplifier? There is also the settling in period which provides information on the implementation.

I found the theory description to be very clear and useful. I would have put it earlier since to me the steady state description in the modified Bloch equations provides a concise description of the dynamics, but I am not suggesting any changes.

Finally, a tiny point that I have never understood what precisely is meant by a “piercing solenoid.” I don’t find the description in their ref 33 to be clear. Is it just a solenoid that goes through the magnetic shield, or ...? Might the authors take the opportunity to clarify this?

Reviewer #3

(Remarks to the Author)

There is no doubt that the article describes an interesting physical concept of a low-field J -oscillator. Nuclear spin order is continuously provided through parahydrogen gas, coupled to the nuclear spin system of interest using the SABRE technique, and electronic feedback is used to generate a continuous oscillating magnetization whose frequency is closely related to the J -couplings in the solution-state nuclear spin system. The results are interesting, although not particularly surprising: NMR feedback oscillations have been observed almost since the birth of NMR in the 1940’s, with particularly extensive investigations by researchers such as the Zürich group of Blum and co-workers in the 1970’s-1980’s (not cited in the article).

The authors claim that the J -oscillator described here has “exhibits superior long-term frequency stability .. in comparison to

conventional NMR lasers". That might be true but I don't see that the evidence for "improved long-term frequency stability" anywhere in the reported results. Since the oscillator is a combination of electronics and nuclear spin dynamics, one would expect the electronic parameters, performance, and noise figure, to play a role both in the oscillation frequency and also its stability. But those possible issues do not seem to be considered at all.

Figure 4 is particularly instructive. It shows the Fourier transform of the RASER signal for mixtures of compounds as a function of electronic feedback gain. The frequencies shift with a change in feedback gain, which reinforces the point made above. In the accompanying text, the authors write that "In the sample that contained 100 % [15N]-pyridine and 50 % 4-amino-[15N]-pyridine, the right peak is about twice as strong as the left one, giving the expected 2:1 ratio that reflects upon its isotopic composition". After some trouble I can identify the spectra in Fig.4 that the authors refer to, and I can also see that the peak amplitudes are roughly in the indicated ratio - although there is no quantitation given, and this ratio looks as if it shifts around depending on the feedback gain (glossed over by the authors). The authors then say "The opposite 1:2 intensity ratio is observed when the enrichment levels are swapped". However, when I try to find the corresponding spectra "when the enrichment levels are swapped", I do not see a swap of the 1:2 intensity ratio. Instead it looks to me as if one peak is simply absent in that case. Am I missing something or are the authors reporting what they would like to see, rather than what is actually observed? None of this inspires confidence for the putative analytical applications advanced by the authors.

The authors report that in one case the Fourier transform of the continuous oscillation gives a linewidth of 337 microHz, whereas the ordinary linewidth without feedback, and limited by spin relaxation, is 37 mHz. From this they conclude that the resolution has been enhanced. However, I would like to know how reproducible is the oscillation frequency? If the oscillations are restarted, is the same frequency observed, within a standard deviation of 337 microHz? Or does it jitter within a standard deviation of 37 mHz (which is what I would expect)? Such statistics are not reported. Nevertheless statements are given which suggest that authors expect applications based on a greatly improved frequency resolution. I do not think that such statements are warranted based on the data presented so far.

The conclusion section on page 12 reads more-or-less like a marketing document. The future is rosy. There is no discussion of limitations, issues that need resolving, or possible obstacles. In short, the presentation is quite uninformative and unscientific.

In summary:

* What are the noteworthy results?

-> demonstration of feedback oscillations for nuclear spins in very low magnetic field

* Will the work be of significance to the field and related fields? How does it compare to the established literature? If the work is not original, please provide relevant references.

-> The feedback device and observations are original and interesting. I have doubts that the observations are significant, but that is a very subjective view.

* Does the work support the conclusions and claims, or is additional evidence needed?

-> In my view the work does not support some of the claims of increased stability, resolution, etc.

* Are there any flaws in the data analysis, interpretation and conclusions? Do these prohibit publication or require revision?

-> There are some flaws in the conclusions and discussion of the results, as detailed above.

* Is the methodology sound? Does the work meet the expected standards in your field?

-> the experimental methodology is sound and meets high standards. However some of the claims made are unsupported and the presentation is uncritical and unscientific in some places

Is there enough detail provided in the methods for the work to be reproduced?

-> yes, I think so.

Version 1:

Reviewer comments:

Reviewer #1

(Remarks to the Author)

The authors have addressed my previous concerns and made changes to the manuscript accordingly. The current version is noticeably written with improved clarity. I have no further comments and recommend accepting this manuscript.

(Remarks on code availability)

Reviewer #3

(Remarks to the Author)

The resubmitted manuscript is more detailed and has less marketing-style hype and more rigorous content than the original submission. However, in my view, the improved content has made it more clear that, although the described experiments are fascinating and are of much interest, some of the claimed achievements are not supported by the evidence. In particular, the abstract makes two claims, of relevance to analytical applications:

- (1) [The advance] “may facilitate precision measurements of J-coupling constants” and
(2) “allows distinguishing mixtures of molecules whose zero-field NMR spectra would otherwise be hard to separate.”

With respect to (1), more detail on the stability and reproducibility of the oscillations is now given in figure 2. This study does support their contention of good reproducibility. However their abstract claim of “precision measurement of J-coupling constants” is undermined by their admission that “A systematic milli-hertz-level offset in the J-oscillation frequency [...] was observed [...] A detailed theoretical analysis of this offset will be presented elsewhere.” This unexplained frequency deviation is evident in the figure. In other words, the authors have no idea, and no evidence, as to whether the observed oscillation frequency, however stable it is, does provide an authentic measurement of the J-coupling, or not. The authors cite some theory of the observed frequency deviation which “will be presented elsewhere”. In the absence of this, there is no evidence at all that the device could be used for highly precise J-coupling measurements.

With respect to (2), I do not see any evidence in the manuscript of distinguishing mixtures of molecules, in cases where the resolution of peaks is not already achieved by conventional means. Indeed, it looks to me as if the introduced feedback simply magnifies the largest of the already resolved peaks - which is exactly as one would expect by non-linear feedback amplification. The authors seem to have believed their own hype in claiming that the observation of one feedback-narrowed peak translates into a real and useful gain in spectral resolution - which would require that unresolved peaks become resolved somehow. That does not happen.

I think that the authors should step back completely from their overhyped claims of enhanced useful spectral resolution. I think that the work is of sufficient wide interest to publish in Nature Communications, but only if the unrealistic claims of possible analytical applications are stripped out or highly qualified.

(Remarks on code availability)

Version 2:

Reviewer comments:

Reviewer #3

(Remarks to the Author)

This second resubmission has further improved the manuscript. It is now clearer and the supporting evidence for the claims made is now clearly laid out. I am still not completely convinced myself by the claims of enhanced resolution, etc., but the main point is that the arguments are now made clearly, without reference to material that is not presented, and free from unsupported marketing claims, and with supporting evidence in plain sight, so that the scientific community is able to debate the issues and challenge the conclusions, if necessary. I judge that these criteria have now been fulfilled and that the paper should be published.

The only very small change I suggest is that when the figure captions states that a spectrum “has been scaled by a factor of 10” the wording should be changed to “has been scaled **up** by a factor of 10” (assuming that this is the correct interpretation).

(Remarks on code availability)

REVIEWER COMMENTS

Reviewer #1 (Remarks to the Author):

See attached review in PDF format. I expressed some criticism and suggested changes to improve the manuscript. Overall, I recommend accepting the manuscript for publication after minor revisions.

Authors' response: We appreciate the referees' constructive criticism and recommendation to accept our manuscript for publication after a minor revision.

The authors demonstrate an intriguing approach to stimulate nuclear magnetic oscillations at zero field via an external magnetic feedback loop and nuclear hyperpolarization. These oscillations exhibit exceptionally long coherence times and ultra-low frequencies associated with the intrinsic J-couplings within the excited molecule. The proposed method is an ultra-low-frequency analogy of the RASER effect, originally observed at high magnetic fields. An interesting novelty here is the capability to recreate “radiation damping” – a key feedback phenomenon for the high-frequency RASER. The authors effectively mimic this by returning the phase-shifted signal back to the sample at various gains using a low-Q solenoid coil. By tuning the gain and the phase shift, the authors have managed to excite multiple RASER frequencies in a variety of ¹⁵N-labelled compounds.

This work is well written, provides sufficient evidence, and has the potential to lead to further investigations of the effect. However, I am not fully convinced by the proposed applications and find a few items questionable:

1) Time-crystals. Despite the catchy name, it is unclear how time-crystal behaviour would be derived from J -oscillations. They disappear immediately after the feedback is turned off as well as match the external feedback frequency. Therefore, authors should provide stronger arguments how time-crystals would manifest using J -oscillations in the liquid-state.

Authors' response: We thank the reviewer for this comment and agree that our initial wording may have overstated the connection to time-crystal; we have accordingly softened the wording in the revised manuscript. Our intention is not to claim the direct realization of a time crystal, but rather to present a versatile platform for studying nonlinear dynamical phenomena in liquid-state molecular spin systems, which can include time-crystalline-like behavior under certain conditions. For instance, when a bias field is applied, the system exhibits distinct dynamical regimes as a function of the applied field and feedback gain, such as fixed points, limit cycles, and chaotic behavior [43]. In the regime of fixed points, the oscillation frequency remains locked to the specific values. Similar behavior can also happen under non-linear feedback schemes.

Changes made to manuscript: the abstract was updated to remove the mentioning of time crystals. The only sentence mentioning this topic now reads as follows "These capabilities facilitate controlled studies of complex phenomena, ranging from chaos to the time crystalline-like behavior in liquid-state molecular systems."

2) J -coupling determination. The proposed method can measure J -oscillations with high precision but the absolute value drifts depending on external gain, sample composition. Authors suggest, however, that numerical simulation may help in such cases but do not provide a confidence limit for the determined J -coupling values. An additional table with the determined J -coupling values and errors would be highly recommended.

Authors' response: We appreciate the referee's constructive criticism. We have now performed additional experiments, substantially expanded the discussion and added a new Figure (Fig. 2). As shown for $[^{15}\text{N}]$ -acetonitrile, the extracted J -coupling exhibits a gradual drift (blue traces in Fig. 2D–F) which we attribute primarily to slow changes in sample composition, most likely due to solvent evaporation during continuous bubbling, leading to subtle variations in exchange dynamics with the SABRE complex. Despite this drift, the J -oscillator yields highly stable and reproducible frequencies, remaining well within 300 μHz across the three independent samples for the same run number, where the sample composition is similar. Since the values of J -couplings for other molecules are expected to drift as well, we do not aim to provide a table with extracted values at this stage. Additionally, the J -oscillators indeed exhibit a systematic frequency offset compared to the J -coupling values extracted from the conventional zero-field measurements. This offset depends on the feedback gain and delay parameters used in the feedback loop. A detailed quantitative analysis of this dependence will be addressed in our future work.

Changes made to manuscript: Multiple changes throughout the text and a new figure added containing new experimental data.

3) Analysis of mixtures. Although the example on a binary mixture is demonstrated, it is also evident that the spin-dynamics and observed frequency is a function of the feedback.

Authors' response: we agree as we also made this observation.

Furthermore, as hyperpolarization by SABRE is only applicable to molecules with good affinity to the catalyst it may also limit the possible scope of detectable molecules. Authors probably anticipate these challenges but should warn the reader.

Author's response: Indeed, this work fundamentally relies on SABRE hyperpolarization to achieve the required population inversion. While this dependence limits the current method to molecules with good affinity to the SABRE catalyst, the scope can be extended in the future by employing alternative hyperpolarization techniques such as SABRE-relay, Overhauser-DNP, or even thermally polarized systems (e.g., delivered via liquid flow). It is worth noting that zero-field NMR as a field has evolved from its origins employing hyperpolarization methods such as PHIP and SABRE (alongside developments using thermal polarization) and continues to expand to encompass all available hyperpolarization strategies.

Changes made to manuscript: We added the sentences in the section 2.5

“Additionally, since the method relies on SABRE hyperpolarization, it is only applicable to molecules with sufficient affinity to the SABRE catalyst, thereby restricting the range of detectable species. Nonetheless, this limitation can be mitigated in the future by employing alternative hyperpolarization schemes such as SABRE-relay, Overhauser-DNP, or even thermally polarized samples under continuous flow, to broaden the scope of molecules accessible to this method.”

Other comments:

4) Conventional RASER. Authors refer to few articles on the RASER effect but it is hard to understand what falls under the “convention”. I suggest using nomenclature of high- and ultra-low frequency RASER which is less ambiguous.

Authors' response: We appreciate the reviewer's suggestion. However, we prefer to use our terminology, J -oscillators, rather than adopting or extending the “RASER” nomenclature. The term RASER (radio-frequency amplification by stimulated emission of radiation) imply radiation processes, while virtually all NMR systems operate in the near-field regime without radiation propagation. Thus, using laser-like terminology is scientifically inaccurate in this context. We nevertheless clearly acknowledge the relation between J -oscillators and previously reported RASER phenomena in the

manuscript. By conventional we mean reported RASER systems relying on Zeeman transitions.

5) Abstract. Bold formatting of “J” should be removed.

Authors’ response: The bold formatting of “J” in the abstract has been removed as suggested.

6) References 9, 12, 28, 30, 37, 38, 39, 43 are missing bibliographic information or have formatting errors. Please fix.

Authors’ response: The References has been checked and properly formatted.

7) Figure A4. It seems that 4 energy levels are missing in $K = 1/2$ space. I suggest marking all levels in $K = 1/2$ space with double lines for clarity.

Authors’ response: We have now added double lines in $K=1/2$ space in the now Figure A9.

8) Figure A11, A inset. It seems that the red bar is upside down compared to figure 4.

Authors’ response: The red bar in Figures A10 and A11 has been corrected to ensure consistency with now Figure 5.

9) Authors have used a spherical sample geometry presumably to ensure uniform magnetic field flux inside the sample. However, during the bubbling with hydrogen gas, the sample geometry could be distorted, especially at flow rates much higher than 20 mL/min. The explanation for setting such low flow rate would clarify the text.

Authors’ response: In our setup, the spherical region is filled up to around 90%, and the liquid remains effectively confined within this volume during continuous bubbling up to 100 scc/m. Even at higher flow rates, where the liquid is extruded to the cylindrical part of the sample, the homogeneous field region of the solenoid is sufficiently large to avoid any field gradients across the sample. A relatively low flow rate of 20 scc/m was intentionally used to minimize liquid evaporation.

Changes made to manuscript: We have added a sentence to the Methods section: “A low flow rate was chosen to slow down sample evaporation.”

10) Algorithm 1. The sequential code provided implies that while reading a portion of the incoming signal NI9263 card cannot output (write) any signal to the solenoid. Does that mean that the external drive is windowed and not continuous? The size of NC should be provided as well.

Authors’ response: We thank the reviewer for the comment. The feedback loop operates in a continuous manner. Both the analog output (AO) and the analog input (AI) modules are synchronized. During operation, new samples are acquired, processed,

and written to the AO card buffer while output from the current buffer is still ongoing, ensuring uninterrupted signal generation.

Changes made to the manuscript: We have added a detailed discussion in Section 4.4, including explicit buffer size requirements and some examples in the revised text.

11) Page 11, “In contrast, high-field lasers lack this flexibility, as their modes spacing cannot be easily tuned (e.g. intrinsic J-couplings are fixed)”. This is a confusing comment since J-couplings are intrinsically fixed for zero-field laser studied as well. Please rewrite.

Authors’ response: We thank the reviewer for the comment.

Changes made to the manuscript: We have revised the text and now it reads:

“...this can be achieved by applying a static bias field, which splits a single J -transition into multiple transitions and allows continuous tuning of their frequency separations simply by adjusting the applied field (to be published elsewhere). Such tunability can enable the systematic exploration of phase transitions in the system’s dynamics—from mode collapse, to frequency combs, to chaos—as the bias field is varied. In contrast, high-field lasers oscillate at the nuclear Larmor frequency, and additional modes arise from intrinsic J -coupled multiplets, these spacings are set by molecular parameters and are therefore not readily tunable.”

12) Equation 7. I recommend modifying HRF notation as it can be confused with standard notation for the Hamiltonian associated with radio-frequency pulses. In addition, changing $B_{Nj}(t)$ to $B_{Sj}(t)$ would add consistency since “S” is used to denote the second spin species already.

Authors’ response: We thank the reviewer for the helpful suggestion.

Changes made to the manuscript: H_{RF} was replaced with H_{fluc} and $B_{Sj}(t)$ was updated to $B_{Kj}(t)$ for consistency.

13) Page 8, 4 paragraph, “(...) significantly improved SNR (...)”. Please specify the SNR achieved by the J-oscillations.

Authors’ response: We thank the reviewer for the comment. We have specified the achieved SNR in the revised text (Sec. 2.5), indicating that the $2J$ oscillations for [^{13}C]-pyruvate exhibit an SNR of approximately 70.

Despite my expressed criticism, I find this work acceptable for publication after minor revisions.

Reviewer #2 (Remarks to the Author):

The authors report a novel and useful approach to generating a steady state signal that is insensitive to temporal variations in the Zeeman interaction. They base their work on driving the scalar coupling in zero magnetic field. The work is well described and could be published as is, however there are a few points they may wish to consider.

Authors' response: We thank the referee for the proposal to publish our work as is.

There are a few places where a steady state response is somewhat confused with a coherence. An example is where a linewidth is reported for a steady state response. Since this is steady state, the response will appear for all time (provided that the spin polarizer continues to function). I would change the language and make clear that the reported linewidth is artificial and is not an indication of a sensor sensitivity.

Authors' response: We thank the reviewer for this comment. We agree that the observed signal corresponds to something different from a conventional free-decay coherence. However, what we call a “dynamic steady state” corresponds to a self-sustained coherence generated under external feedback, rather than being an artificial feature.

As discussed in Section 2.2 and shown in Fig. 2D–F, the J -oscillator reproducibly returns to the same frequency response upon each restart. The J -oscillation frequency exhibits a small, systematic offset relative to the value measured with conventional zero-field NMR. This offset is stable and reproducible across repeated runs and among three independently prepared samples, demonstrating that it strongly correlates with the measured J -coupling of the sample. In fact, the linewidth narrowing does not persist indefinitely with measurement time, as slow drifts in the frequency of a running J -oscillator are also observed (see Fig. A2A). We have clarified these points in the revised text.

Changes made to the manuscript: multiple sentences in the text and the SI.

The same factor makes the use of the method for time-crystals somewhat less interesting.

Authors' response: We thank the reviewer for the comment. As addressed in our response to Reviewer #1 above, we do not claim direct observation of a time crystal. In fact, we mean that nonlinear spin dynamics of liquid-state molecular spin systems can exhibit time-crystalline-like behavior.

Changes made to the manuscript: mentioning time crystals was removed from the abstract.

The use of an amplifier in the feedback loop to drive the response is interesting, though it has been done before including by Laukien (to reduce radiation damping). Is there any

analysis on the effects of noise from the addition of the amplifier? There is also the settling-in period which provides information on the implementation.

Authors' response: We appreciate the reviewer's comment. In our system, the piercing solenoid configuration is designed so that the solenoid field is localized inside and produces no field at the OPM sensor location. Therefore, any noise generated by the amplifier and fed into the solenoid will not couple to the OPM sensor. To verify this, we measured the OPM's noise floor as we varied the external feedback gain: the noise floor remains essentially unchanged with increasing gain (see Fig. A5, A6, and updated discussion in Section 4.4).

Now we also provide a demonstration in Fig. 1E, showing a zoomed-in view of the settling-in period (where only background OPM noise under the active feedback is present) before the J -oscillation starts to emerge. Details of the feedback initiation have been added to Section 4.4. Specifically, at the start of the feedback loop, we assume that the prior read and write data samples are zero, consistent with the fact that no detectable signal is initially present.

Changes made to the manuscript: In addition to the new data and figures added to the text, references 32-33 for the effect of external feedback on the coil Q-factor have been cited.

I found the theory description to be very clear and useful. I would have put it earlier since to me the steady state description in the modified Bloch equations provides a concise description of the dynamics, but I am not suggesting any changes.

Authors' response: We appreciate finding the theory section useful.

Changes made to the manuscript: none.

Finally, a tiny point that I have never understood what precisely is meant by a "piercing solenoid." I don't find the description in their ref 33 to be clear. Is it just a solenoid that goes through the magnetic shield, or ...? Might the authors take the opportunity to clarify this?

Authors' response: "Piercing" means that a solenoid goes through the magnetic shield. Your understanding is exactly right and we explain it better in the text now. Such solenoid is specifically designed to generate a magnetic field at the sample position while producing (ideally) no field at the OPM sensor. We have also updated the discussion in Section 4.4 to include the algorithm used to mitigate the effect of any small residual field that may leak to the OPM.

Reviewer #3 (Remarks to the Author):

There is no doubt that the article describes an interesting physical concept of a low-field J -oscillator. Nuclear spin order is continuously provided through parahydrogen gas, coupled to the nuclear spin system of interest using the SABRE technique, and electronic feedback is used to generate a continuous oscillating magnetization whose frequency is closely related to the J -couplings in the solution-state nuclear spin system. The results are interesting, although not particularly surprising: NMR feedback oscillations have been observed almost since the birth of NMR in the 1940's, with particularly extensive investigations by researchers such as the Zürich group of Blum and co-workers in the 1970's-1980's (not cited in the article).

Authors' response: more likely, the reviewer meant the works of Brum (not Blum) and we cited the following works (Refs. [16-17]).

The authors claim that the J -oscillator described here has “exhibits superior long-term frequency stability .. in comparison to conventional NMR lasers”. That might be true but I don't see that the evidence for “improved long-term frequency stability” anywhere in the reported results.

Authors' response: We thank the reviewer for this insightful comment, and we have now clarified the relevant points in the revised manuscript.

In contrast to high-field NMR lasers, whose self-oscillation frequencies depend on nuclear Zeeman transitions and are therefore highly sensitive to the stability of the external magnetic field (e.g., temperature-driven drifts of the magnet), the J -oscillators operate on molecular J -coupling constants, which are intrinsic molecular properties and therefore exhibit much better frequency stability. Compared to Fig. 1H, high-field lasers show substantially greater frequency fluctuations under much shorter monitoring durations (see Fig. 2 and 3 in [40] Fleischer et al., *Approaching the Ultimate Limit in Measurement Precision with RASER NMR*, *Appl. Magn. Reson.*, 2023).

Since the oscillator is a combination of electronics and nuclear spin dynamics, one would expect the electronic parameters, performance, and noise figure, to play a role both in the oscillation frequency and also its stability. But those possible issues do not seem to be considered at all.

Authors' response: the feedback gain and delay parameters indeed shift the J -oscillation frequency albeit in a predictable manner. The two parameters are digitally controlled, ensuring high robustness against analog noise in the feedback loop.

The main bottleneck of the performance of the J -oscillator is the piercing solenoid leakage field which couples the amplifier noise into the feedback loop. We have updated Section 4.4 to discuss and address this issue and revised description of the Algorithm 1 to include the mitigation procedure. This has been experimentally verified — e.g., Figs. A5 and A6 demonstrate that the OPM noise floor remains unchanged with

increasing feedback gain. Other problems like additional signal propagation delays in the feedback loop are also addressed in Section 4.4.

Finally, the stability and reproducibility of the J -oscillator have been systematically tested using three independently prepared, identical samples of naturally abundant acetonitrile. Each sample was measured repeatedly (11 runs per sample), and in every case the J -oscillator frequency reproducibly tracked the gradual increase of the J -coupling constant with much smaller fluctuations compared to the corresponding free-decay (FD) measurements (see Fig. 2D–2F). The small systematic offset between the J -oscillation and the J -coupling value extracted from the FD measurements is stable and reproducible across all runs and samples, confirming the high long-term frequency stability of the J -oscillator. A detailed theoretical analysis of this offset will be presented in future work.

Figure 4 is particularly instructive. It shows the Fourier transform of the RASER signal for mixtures of compounds as a function of electronic feedback gain. The frequencies shift with a change in feedback gain, which reinforces the point made above. In the accompanying text, the authors write that “In the sample that contained 100 % [15N]-pyridine and 50 % 4-amino-[15N]-pyridine, the right peak is about twice as strong as the left one, giving the expected 2:1 ratio that reflects upon its isotopic composition”. After some trouble I can identify the spectra in Fig.4 that the authors refer to, and I can also see that the peak amplitudes are roughly in the indicated ratio - although there is no quantitation given, and this ratio looks as if it shifts around depending on the feedback gain (glossed over by the authors). The authors then say “The opposite 1:2 intensity ratio is observed when the enrichment levels are swapped”. However, when I try to find the corresponding spectra “when the enrichment levels are swapped”, I do not see a swap of the 1:2 intensity ratio. Instead it looks to me as if one peak is simply absent in that case. Am I missing something or are the authors reporting what they would like to see, rather than what is actually observed? None of this inspires confidence for the putative analytical applications advanced by the authors.

Authors' response: we now make the text clearer and provide a reference to the SI (Fig. A16) where we indeed see the opposite ratio with larger applied gains.

The authors report that in one case the Fourier transform of the continuous oscillation gives a linewidth of 337 microHz, whereas the ordinary linewidth without feedback, and limited by spin relaxation, is 37 mHz. From this they conclude that the resolution has been enhanced. However, I would like to know how reproducible is the oscillation frequency? If the oscillations are restarted, is the same frequency observed, within a standard deviation of 337 microHz? Or does it jitter within a standard deviation of 37 mHz (which is what I would expect)? Such statistics are not reported. Nevertheless statements are given which suggest that authors expect applications based on a greatly improved frequency resolution. I do not think that such statements are warranted based on the data presented so far.

Authors' response: We thank the reviewer for raising this question. We have now clarified this point and added relevant discussion in the revised manuscript (Section 2.2). Unlike systems whose linewidth is dominated by spontaneous emission (the Schawlow–Townes limit), the J -oscillator frequencies show excellent reproducibility upon repeated restarts. As shown in now Fig. 2D–2F, the frequency offset between the J -oscillator and the value measured by a conventional zero-field NMR is also stable and reproducible across repeated runs and independently prepared samples. When comparing the same n -th run among three identical acetonitrile samples, the measured oscillation frequencies agree within 300 μHz , confirming that the frequency reproducibility is comparable to the observed linewidth and far better than the 32-37 mHz linewidths (naturally abundant sample has a slightly narrower linewidth) of the free-decay signal.

The conclusion section on page 12 reads more-or-less like a marketing document. The future is rosy. There is no discussion of limitations, issues that need resolving, or possible obstacles. In short, the presentation is quite uninformative and unscientific.

Authors' response: We now deleted the conclusions section as per the journal's requirement but clearly highlighted open questions in each subsection.

Changes made to the manuscript: Multiple changes in the text to discuss limitations/open questions.

In summary:

* What are the noteworthy results? ->

demonstration of feedback oscillations for nuclear spins in very low magnetic field

* Will the work be of significance to the field and related fields? How does it compare to the established literature? If the work is not original, please provide relevant references.->

The feedback device and observations are original and interesting. I have doubts that the observations are significant, but that is a very subjective view.

* Does the work support the conclusions and claims, or is additional evidence needed?-> In my view the work does not support some of the claims of increased stability, resolution, etc.

Authors' response: Updated manuscript clearly supports claims made about stability and resolution. Additionally, we clearly demonstrate that the ratio of the observed J -oscillator peak intensities in the case of molecular mixture corresponds to the proportion of chemicals (Fig. 5 and Fig. A16).

* Are there any flaws in the data analysis, interpretation and conclusions? Do these prohibit publication or require revision?->

There are some flaws in the conclusions and discussion of the results, as detailed above.

Authors' response: See above.

* Is the methodology sound? Does the work meet the expected standards in your field?->

the experimental methodology is sound and meets high standards. However some of the claims made are unsupported and the presentation is uncritical and unscientific in some places

Authors' response: Updated manuscript fully supports claims that have been made.

Is there enough detail provided in the methods for the work to be reproduced?-> yes, I think so.

REVIEWER COMMENTS

Reviewer #1 (Remarks to the Author):

The authors have addressed my previous concerns and made changes to the manuscript accordingly. The current version is noticeably written with improved clarity. I have no further comments and recommend accepting this manuscript.

Reviewer #3 (Remarks to the Author):

The resubmitted manuscript is more detailed and has less marketing-style hype and more rigorous content than the original submission. However, in my view, the improved content has made it more clear that, although the **described experiments are fascinating and are of much interest**, some of the claimed achievements are not supported by the evidence. In particular, the abstract makes two claims, of relevance to analytical applications:

- (1) [The advance] “may facilitate precision measurements of J-coupling constants” and
- (2) “allows distinguishing mixtures of molecules whose zero-field NMR spectra would otherwise be hard to separate.”

With respect to (1), more detail on the stability and reproducibility of the oscillations is now given in figure 2. This study does support their contention of good reproducibility. However their abstract claim of “precision measurement of J-coupling constants” **is undermined by their admission** that “A systematic milli-hertz-level offset in the J-oscillation frequency [...] was observed [...] A detailed theoretical analysis of this offset will be presented elsewhere.” This unexplained frequency deviation is evident in the figure. In other words, the authors have no idea, and no evidence, as to whether the observed oscillation frequency, however stable it is, does provide an authentic measurement of the J-coupling, or not. The authors cite some theory of the observed frequency deviation which “will be presented elsewhere”. In the absence of this, there is no evidence at all that the device could be used for highly precise J-coupling measurements.

The authors’ response: As discussed in the manuscript, frequency of the *J*-oscillators is determined by several factors. It depends on the parameters of the feedback loop as well as intrinsic spin-dynamical properties of the sample, including *J*-couplings and the relaxation time scales of the relevant transitions. Crucially, the developed theory shows (to be published elsewhere), and numerical simulations confirm, that the oscillation frequency is independent of the feedback gain and, therefore, independent of the signal intensity or polarization level of the sample, see the Supplementary Information for review only. A simple calibration procedure allows the *J*-coupling constant to be extracted from measurements performed at several feedback delays.

Our intention is not to withhold this theory, but to present it in a dedicated publication where the full derivation and validation will be given. To avoid overstating the present

results, we therefore adopted cautious wording in the abstract (e.g., “This MAY facilitate precision measurements of J -coupling constants ...”) when referring to precision measurements.

Changes made to the manuscript: None.

With respect to (2), **I do not see any evidence** in the manuscript of distinguishing mixtures of molecules, in cases where the resolution of peaks is not already achieved by conventional means. Indeed, it looks to me as if the introduced feedback simply magnifies the largest of the already resolved peaks - which is exactly as one would expect by non-linear feedback amplification. The authors seem to have believed their own hype in claiming that the observation of one feedback-narrowed peak translates into a real and useful gain in spectral resolution - which would require that unresolved peaks become resolved somehow. That does not happen.

I think that the authors should step back completely from their overhyped claims of enhanced useful spectral resolution. I think that the work is of sufficient wide interest to publish in Nature Communications, but only if the unrealistic claims of possible analytical applications are stripped out or highly qualified.

The authors' response: To clarify our claims regarding mixture analysis, we have revised Fig. 5 and Fig. A16 to include the corresponding conventional zero-field NMR spectra, shown as shaded traces for direct visual comparison. In both figures, the conventional spectra are broad, with overlapping features that do not provide sufficient resolution to distinguish the individual components of the mixture. In contrast, the J -oscillator method yields narrow, well-separated oscillation features whose identities can be assigned even when the zero-field spectra lack adequate clarity.

We have also added Fig. A17 which demonstrates that adjusting the feedback-delay parameter provides an additional degree of control: by selecting appropriate delays, the J -oscillation peaks associated with individual complexes can be selectively excited within the same sample. This tunability further assists in disentangling contributions from structurally similar molecules.

Changes made to the manuscript:

1. Fig. 5 and Fig. A16 have been updated to include the corresponding conventional zero-field NMR spectra for direct visual comparison.
2. A new Fig. A17 has been added to demonstrate how adjusting the feedback-delay parameter provides additional control for mixture analysis.
3. The relevant sentence in the manuscript has been revised to read:
“The results suggest that the J -oscillator method can assist in analyzing complex mixtures containing structurally similar molecules, particularly in situations where conventional zero-field NMR does not provide sufficient resolution to distinguish signals [24].”

Reviewer #3 (Remarks to the Author):

This second resubmission has further improved the manuscript. It is now clearer and the supporting evidence for the claims made is now clearly laid out. I am still not completely convinced myself by the claims of enhanced resolution, etc., but the main point is that the arguments are now made clearly, without reference to material that is not presented, and free from unsupported marketing claims, and with supporting evidence in plain sight, so that the scientific community is able to debate the issues and challenge the conclusions, if necessary. I judge that these criteria have now been fulfilled and that the paper should be published.

The authors' response: We thank the reviewer for their positive reassessment.

The only very small change I suggest is that when the figure captions states that a spectrum "has been scaled by a factor of 10" the wording should be changed to "has been scaled **up** by a factor of 10" (assuming that this is the correct interpretation).

The authors' response: We have revised all relevant figure captions to read "**scaled up by a factor of 10**".

Review of “Quantum Magnetic J -Oscillators” by Jingyan Xu, Raphael Kircher, Oleg Tretiak, Dmitry Budker and Danila A. Barskiy

The authors demonstrate an intriguing approach to stimulate nuclear magnetic oscillations at zero field via an external magnetic feedback loop and nuclear hyperpolarization. These oscillations exhibit exceptionally long coherence times and ultra-low frequencies associated with the intrinsic J -couplings within the excited molecule. The proposed method is an ultra-low-frequency analogy of the RASER effect, originally observed at high magnetic fields. An interesting novelty here is the capability to recreate “radiation damping” – a key feedback phenomenon for the high-frequency RASER. The authors effectively mimic this by returning the phase-shifted signal back to the sample at various gains using a low-Q solenoid coil. By tuning the gain and the phase shift, the authors have managed to excite multiple RASER frequencies in a variety of ^{15}N -labelled compounds.

This work is well written, provides sufficient evidence, and has the potential to lead to further investigations of the effect. However, I am not fully convinced by the proposed applications and find a few items questionable:

1) Time-crystals. Despite the catchy name, it is unclear how time-crystal behaviour would be derived from J -oscillations. They disappear immediately after the feedback is turned off as well as match the external feedback frequency. Therefore, authors should provide stronger arguments how time-crystals would manifest using J -oscillations in the liquid-state.

2) J -coupling determination. The proposed method can measure J -oscillations with high precision but the absolute value drifts depending on external gain, sample composition. Authors suggest, however, that numerical simulation may help in such cases but do not provide a confidence limit for the determined J -coupling values. An additional table with the determined J -coupling values and errors would be highly recommended.

3) Analysis of mixtures. Although the example on a binary mixture is demonstrated, it is also evident that the spin-dynamics and observed frequency is a function of the feedback. Furthermore, as hyperpolarization by SABRE is only applicable to molecules with good affinity to the catalyst it may also limit the possible scope of detectable molecules. Authors probably anticipate these challenges but should warn the reader.

Other comments:

4) Conventional RASER. Authors refer to few articles on the RASER effect but it is hard to understand what falls under the “convention”. I suggest using nomenclature of high- and ultra-low frequency RASER which is less ambiguous.

5) Abstract. Bold formatting of “ \mathcal{J} ” should be removed.

6) References 9, 12, 28, 30, 37, 38, 39, 43 are missing bibliographic information or have formatting errors. Please fix.

7) Figure A4. It seems that 4 energy levels are missing in $K = 1/2$ space. I suggest marking all levels in $K = 1/2$ space with double lines for clarity.

8) Figure A11, A inset. It seems that the red bar is upside down compared to figure 4.

9) Authors have used a spherical sample geometry presumably to ensure uniform magnetic field flux inside the sample. However, during the bubbling with hydrogen gas, the sample geometry could be distorted, especially at flow rates much higher than 20 mL/min. The explanation for setting such low flow rate would clarify the text.

10) Algorithm 1. The sequential code provided implies that while reading a portion of the incoming signal NI9263 card cannot output (write) any signal to the solenoid. Does that mean that the external drive is windowed and not continuous? The size of N_C should be provided as well.

11) Page 11, “In contrast, high-field lasers lack this flexibility, as their modes spacing cannot be easily tuned (e.g. intrinsic J -couplings are fixed)”. This is a confusing comment since J -couplings are intrinsically fixed for zero-field laser studied as well. Please rewrite.

12) Equation 7. I recommend modifying H_{RF} notation as it can be confused with standard notation for the Hamiltonian associated with radio-frequency pulses. In addition, changing $B_{Nj}(t)$ to $B_{Sj}(t)$ would add consistency since “S” is used to denote the second spin species already.

13) Page 8, 4 paragraph, “(...) significantly improved SNR (...)”. Please specify the SNR achieved by the J -oscillations.

Despite my expressed criticism, I find this work **acceptable** for publication after **minor revisions**.